# Heterogeneity of Increases in Net Primary Production under Intensified Human Activity and Climate Variability on the Loess Plateau of China

**Xiangnan Ni [1], Wei Guo [1,2,\*], Xiaoting Li [1]**  **and Shuheng Li [3]**

[1]  Department of Earth and Environmental Sciences, Xi'an Jiaotong University, Xi'an 710049, China
[2]  Institution of Global Environmental Change, Xi'an Jiaotong University, Xi'an 710049, China
[3]  College of Urban and Environmental Sciences, Northwest University, Xi'an 710127, China
[\*]  Correspondence: williamguo@xjtu.edu.cn; Tel.: +86-15929301378

**Abstract:** Regrowth of forests is expected to be an important driver in the large uptake of anthropogenic $CO_2$ emissions by the terrestrial biosphere. Yet estimates of carbon sink capacity in mid-high latitude regrowth forests still remain unclear. The Loess Plateau (LP), a key region of the Grain to Green Program (GTGP), leads in the greening of China, while China leads in the greening of the world. For the sake of global ecological sustainability and accurate global carbon sink evaluation, the detection and attribution of vegetation growth on the LP requires further research after 20 years of ecological restoration. In this study, significant continuous rises (increases of 7.45 gC·m$^{-2}$·a$^{-2}$, $R^2$ = 0.9328, $p < 0.01$) in net primary production (NPP) have occurred in the past 20 years. Rapid growth of forest NPP and expansion of forested areas in the southeastern regions has led to vegetation restoration on the LP. Human activities contributed 64.2% to the NPP increases, while climate variations contributed 35.8%. NPP in forests and croplands was dominated by human activities, while grassland NPP was mainly influenced by climate variations on the LP. Meanwhile, a strong El Niño event exacerbated the obstruction of large-scale ecological restoration. These conclusions can provide theoretical support for carbon-cycle assessment and the evaluation of sustainable development.

**Keywords:** net primary production; afforestation; climate change; Grain to Green Program; El Niño–Southern Oscillation

## 1. Introduction

The global carbon cycle plays a critical role in biogeochemical processes, which have important impacts on the nitrogen cycle, the phosphorus cycle, and other material cycles in ecosystems [1–3]. The terrestrial carbon cycle is the most complex part among multiple different processes operating on various temporal and spatial scales and the manifestation of human activities [4–6]. Meanwhile, the terrestrial carbon cycle and global climate change are strongly coupled, something that has been regarded as a frontier subject of global change research [7–9]. Tropical forests contain about 40−50% of the terrestrial carbon stock [10–12]. However, terrestrial carbon sinks are predominantly in mid-to-high latitudes, rather than tropical forests [13,14]. Monitoring and quantifying changes in carbon sinks in mid-to-high latitudes, therefore, is essential for understanding the global carbon cycle and future climate change.

A great global greening trend has been observed during the past four decades, and the greening trend in China is believed to have been greater than that of the global average, especially in the northern semi-arid regions [15–17]. Vegetation on the Loess Plateau (LP) has increased significantly in the past 20 years [18–20]. However, ecological recovery on the LP is not an easy task due to the harsh climatic conditions and environmental factors [21–23]. The LP is the largest loess accumulation region in the world, with a continental monsoon climate, little and concentrated year-round precipitation, frequent human activities, serious

soil erosion, and extremely unstable ecosystems [24–26]. The vegetation on the LP has nevertheless shown significant improvement, which is closely related to China's largest ecological engineering program, the Grain to Green Program (GTGP) [27–29]. This program aims to (1) reclaim croplands back to forests, (2) afforest semiarid mountainous regions, and (3) reinforce forest conservation. As the key region of GTGP, the greening of the LP marks the great success of China's ecological engineering programs and offers a model for global ecological restoration, which also suggests that vegetation in mid-to-high latitudes has a very high carbon sink potential [30,31]. Further studies in the detection and attribution of the rapid growth of vegetation on the LP are needed to provide guidance for global ecological programs.

At the ecosystem scale, the net photosynthetic flux of carbon is referred to as net primary productivity (NPP). As one of the key parameters that characterize terrestrial ecosystems, vegetation NPP can reflect the production capacity of plant communities under various natural environmental conditions, which is an important parameter for in-depth understanding of the surface carbon cycle process [32]. In previous researches, many NPP estimation models of terrestrial ecosystems have been proposed and applied to different regions [33], summarized into climate production potential models, ecological process models and light use efficiency (LUE) models. Of the various models used, land-cover type is a very important input parameter which decides the plant's functional type and photosynthetic efficiency. Uncertainties in previous NPP estimates indicate the many previous models largely ignore the impact of continuous land-use changes. However, high-intensity human activities have caused drastic changes in the land cover of the LP in past decades. Therefore, the influence of continuous land-use changes on model parameters must be considered in order to accurately estimate the NPP and quantify carbon sinks on the LP.

Vegetation variations on the LP have been affected by both climate change and human activities [34,35]. In recent decades, significant climate variations and increasing human activity have occurred on the LP [17,25]. The significant growth of vegetation on the LP is a response to an interplay between climatic changes and human disturbance, which has attracted widespread attention [19,20,36]. A large number of previous studies have explored the mechanisms by which human activities and climate influence vegetation variations on the LP. However, the driving mechanism of vegetation variations is still unclear. Most studies have concluded that strong human activity has dominated the vegetation growth on the LP in recent decades [19,20], although some research has indicated that climate change has played more important roles in regulating vegetation dynamics [36]. Many studies have concluded that precipitation is the most important climatic factor for variation in vegetation in semi-arid areas [25,37]. However, some research has found that temperature also has tangible effects on vegetation variation on the LP [38]. In addition, the impacts of human activities—especially of large-scale ecological engineering programs—on the LP's highly heterogeneous vegetation types and landforms still needed to be evaluated and quantified. To this end, a quantitative analysis of the relative weighting of the impacts of climate change and human activities on vegetation variation would make the attribution of greening more reliable.

Meanwhile, climate variations or extremes caused by atmospheric oscillations should receive more attention in the study of the driving mechanism of vegetation variation. Few previous studies have been performed on El Niño–Southern Oscillation (ENSO) and vegetation growth in China, since this country is not among the core regions influenced by ENSO events [39,40]. The impact of ENSO events on vegetation growth has therefore remained unclear [41]. However, the LP is deeply affected by the monsoon and ENSO events have direct impacts on the monsoon climate [42] which has a profound impact vegetation variation on the LP. Therefore, the impact of ENSO on vegetation variation requires further research in order to guide the implementation of large-scale ecological restoration projects.

This paper focuses on detection and attribution of increases in NPP on the LP from 2001 to 2020 based on multiple variables from satellite sensors and reanalysis datasets. The objectives of this paper are to: (1) optimize the Carnegie–Ames–Stanford approach (CASA) model to accurately simulate NPP on the LP over the past 20 years; (2) study the spatial–temporal patterns of NPP variations on the LP; (3) disentangle the influence of climate change and human activities on vegetation variations; and (4) discuss the impacts of atmospheric oscillation on vegetation NPP on the LP.

## 2. Materials and Methods

### 2.1. Study Area

The LP, located in the north of China, extends from 33°41′N to 41°16′N and 100°52′E to 114°33′E, surrounded by tall mountains, lying to the east of the Riyue Mountains, west of the Taihang Mountains, south of the Yin Mountains, and north of the Qinling Mountains (Figure 1). The total area totals approximately 620,000 km$^2$ over an area 1300 km long and 800 km wide. The administrative divisions include Shanxi Province, Ningxia Hui Autonomous Region, Shaanxi Province, Gansu Province, Qinghai Province, Inner Mongolia Autonomous Region, and Henan Province. The LP is the largest, most concentrated and typical loess landform unit in the world, and is famous for its huge accumulation of loess (30~200 m) and serious soil erosion [35]. Temperate deciduous forest, forest–steppe, typical steppe, and temperate desert steppe appear from southeast to northwest. However, the mountain vegetation in the east and south of the LP show obvious differences in vertical distribution due to changes in temperature and precipitation with altitude. The heterogeneity of climatic conditions and ecological environment highlight the regional differences in the changes to vegetation on the LP. According to the heterogeneity of its climatic conditions and surface characteristics, the LP can be divided into six sub-regions: (1) temperate steppe desert (TSD), (2) alpine forest–steppe (AFS), (3) temperate desert–steppe (TDS), (4) temperate typical steppe (TTS), (5) temperate forest–steppe (TFS), and (6) temperate deciduous forest (TDF). In general, the precipitation and temperature decreases from southeast to northwest (Table 1).

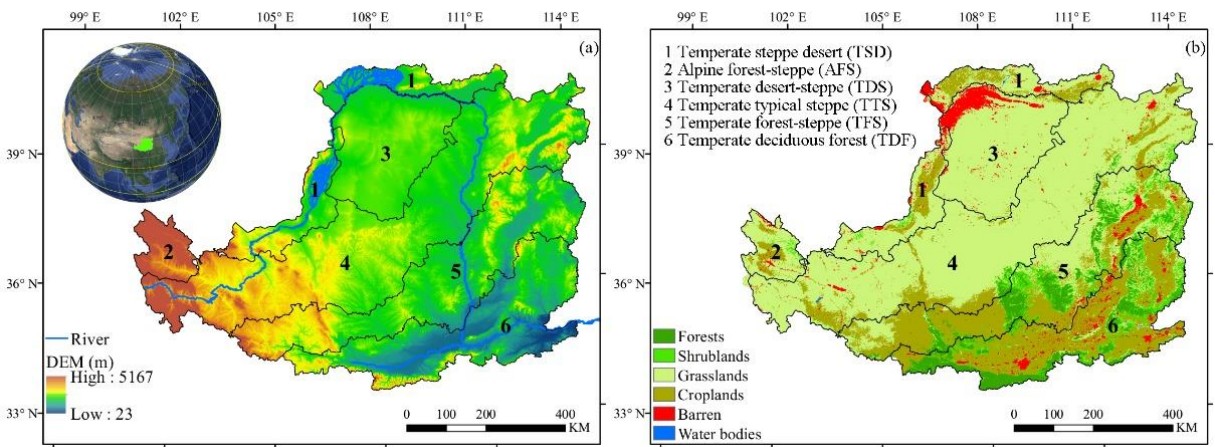

**Figure 1.** Map of location (**a**) and land cover (**b**) of the Loess Plateau in China.

**Table 1.** Climate conditions and vegetation types in the sub-regions in 2020.

| Sub-Regions | Area (km$^2$) | Annual Mean Temperature (°C) | Annual Precipitation (mm) | Vegetation Types |
|---|---|---|---|---|
| TSD | 5.07 | 8.77 | 251.75 | Croplands; steppe–desert |
| AFS | 3.51 | 4.60 | 387.57 | Desert; steppe; forest |
| TDS | 8.06 | 8.33 | 288.94 | Desert; steppe |

**Table 1.** *Cont.*

| Sub-Regions | Area (km²) | Annual Mean Temperature (°C) | Annual Precipitation (mm) | Vegetation Types |
|---|---|---|---|---|
| TTS | 16.80 | 8.17 | 413.53 | Steppe |
| TFS | 13.37 | 10.12 | 519.30 | Forest–steppe |
| TDF | 11.61 | 12.40 | 600.56 | Forests; croplands |

*2.2. Data Used*

Various variables from satellite sensors over the LP and reanalysis datasets were used in this research. These included land-cover maps, the normalized difference vegetation index (NDVI), evapotranspiration (ET) from the Moderate Resolution Imaging Spectroradiometer (MODIS) on the Terra platform, temperature and solar radiation from the Famine Early Warning Systems Network (FEWS NET) Land Data Assimilation System (FLDAS), precipitation from Global Precipitation Measurement (GPM), population distribution from LandScan, the sea surface temperature anomaly (SSTa) index for Niño 3.4 (5°N to 5°S, 170°W to 120°W), and monthly averaged total column water vapor from the ERA5 dataset.

*MODIS land cover dataset.* Collection 6 Terra and Aqua MODIS land cover product (MCD12Q1) from 2001 to 2020 at yearly temporal frequency and 500 m spatial resolution was used to indicate the spatial–temporal patterns of land-use change in our study area [43]. This product provides several classification schemes. The map of the International Geosphere–Biosphere Programme (IGBP) classification scheme was adopted for this research.

*MODIS NDVI datasets.* Collection 6 Terra MODIS NDVI products (MOD13A3) for the period January 2001 to December 2020 were used in this study. The NDVI dataset provides monthly composite NDVI at a 1 km spatial resolution. The NDVI over our study area was refined by removing pixels with shadows, inland snow/ice, inland water, and mixed clouds.

*MODIS ET/PET datasets.* Collection 6 Terra MODIS Evapotranspiration (ET) products (MOD16A2GF) for the period January 2001 to December 2020 were used in this study. The dataset provides year-end gap-filled 8-day composite ET, potential ET (PET) at a 500 m pixel resolution. The ET/PET over our study area was refined by removing pixels with negative values.

*FLDAS temperature and radiation datasets.* FLDAS Noah Land Surface Model L4 dataset (FLDAS_NOAH01_C_GL_M) for the period January 2001 to December 2020 was used in this study. The dataset contains a series of global monthly land surface parameters simulated from the Noah 3.6.1 model at a 0.1° resolution. Near-surface air temperature and surface downward shortwave radiation were selected to indicate the monthly air temperature and solar radiation over the LP.

*GPM IMERG precipitation datasets.* Version 06B Integrated Multi-satellite precipitation products (GPM_3IMERGM) for the period January 2001 to December 2020 were used in this study. The precipitation estimates from the various precipitation-relevant satellite passive microwave (PMW) sensors were used. The dataset provides monthly precipitation over the LP at a 0.1° resolution.

*Population Distribution datasets.* LandScan global population distribution data from 2001 and 2020 were used in this study. LandScan was developed using best available demographic (census) and geographic data, remote sensing, and imagery analysis techniques within a multivariate dasymetric modeling framework. The dataset provides an ambient population distribution (averaged over 24 h) on the LP.

The MODIS land-cover datasets with IGBP classification scheme were reclassified to six land-cover types: forests, shrublands, grasslands, croplands, barren, and water bodies (Figure 1). The 1 km NDVI products, FLDAS data, and precipitation products were spatially aggregated to a resolution of 500m and were then used in our model simulation and analyses.

*2.3. Contribution of Each Driving Factor to Interannual Variation in NPP*

Changes in vegetation NPP are mainly affected by climatic and other factors. Climate factors primarily include temperature, precipitation and solar radiation, while other factors mainly refer to human activities on the LP. In order to further understand the influence weights of human activities and climate factors on changes to vegetation NPP, the partial derivative method [44] was used to calculate the contribution of climate factors and human activity to interannual vegetation growth for each pixel during 2001 to 2020. This was done by using

$$
\begin{aligned}
Slope &= C(tem) + C(rad) + C(pre) + UF \\
&= \left(\frac{\partial NPP}{\partial tem}\right) \times \left(\frac{\partial tem}{\partial n}\right) + \left(\frac{\partial NPP}{\partial rad}\right) \times \left(\frac{\partial rad}{\partial n}\right) + \left(\frac{\partial NPP}{\partial pre}\right) \times \left(\frac{\partial pre}{\partial n}\right) + UF
\end{aligned}
\tag{1}
$$

where *Slope* [gC·m$^{-2}$·a$^{-2}$] is the increase rate of *NPP*; *C* [gC·m$^{-2}$·a$^{-2}$] represents contributions of climate factors to vegetation growth; *UF* is residual error, which represents the contribution of human activities; *tem*, *pre*, and *rad* are annual average temperature, annual precipitation, and annual solar radiation, respectively; and *n* is year.

*2.4. Interactive Effect of Driving Factors on the NPP*

Interaction effects become more and more important in the attribution analysis of vegetation. Geo-detector (i.e., Geographical Detector) is a statistical tool to measure spatial stratified heterogeneity (SSH) and to make attribution for/by SSH [45]. In this study, a geo-detector was used to investigate the interaction effects of explanatory variables on NPP in the various sub-regions of the LP. This was done using Geo-detector software (http://geodetector.cn/ (accessed on 1 January 2022)).

**3. Model and Evaluation**

*3.1. CASA Model*

In this study, the NPP was calculated by Carnegie–Ames–Stanford approach (CASA) model, which is a light use efficiency model based on global satellite and surface data [46]. This was done using

$$
NPP(x,t) = APAR(x,t) \times \varepsilon(x,t)
\tag{2}
$$

where *t* is time, *x* is the pixel number, *NPP* (*x, t*) [gC·m$^{-2}$] is the net primary production of pixel *x* at time *t*, *APAR* (*x, t*) [MJ·m$^{-2}$] represents absorbed photosynthetically active radiation of pixel *x* at time *t*, and *ε* (*x, t*) [gC·MJ$^{-1}$] is the light use efficiency for pixel *x* at time *t*. The absorbed photosynthetically active radiation was calculated as follows:

$$
APAR(x,t) = 0.47 \times SOL(x,t) \times FPAR(x,t)
\tag{3}
$$

where *SOL* (*x, t*) [MJ·m$^{-2}$] is solar radiation of pixel *x* at time *t* from FLDAS radiation datasets, *FPAR* (*x, t*) is the fraction of absorbed photosynthetically active radiation of pixel *x* at time *t*, and the constant 0.47 is the ratio of effective solar radiation that can be used by vegetation to the total solar radiation. The FPAR can be calculated as follows:

$$
FPAR(x,t) = \frac{RVI(x,t) - RVI_{i,\min}}{RVI_{i,\max} - RVI_{i,\min}} \times (FPAR_{\max} - FPAR_{\min}) + FPAR_{\min}
\tag{4}
$$

where $FPAR_{\max} = 0.95$ and $FPAR_{\min} = 0.001$ represent maximum and minimum FPAR, and $RVI_{\max}$ and $RVI_{\min}$ represent maximum and minimum RVI, which are determined by the land-cover type [47] (Table 2). RVI can be calculated from the MODIS NDVI dataset.

$$
RVI(x,t) = \left[\frac{1 + NDVI(x,t)}{1 - NDVI(x,t)}\right]
\tag{5}
$$

The light use efficiency is influenced by many environmental factors, including temperature and moisture. The light use efficiency was calculated by using the model established by Potter and Field [48,49]. The efficiency, $\varepsilon$, is calculated as follows:

$$\varepsilon(x,t) = T_{\varepsilon 1}(x,t) \times T_{\varepsilon 2}(x,t) \times W_{\varepsilon}(x,t) \times \varepsilon_{\max} \tag{6}$$

$$T_{\varepsilon 1}(x,t) = 0.8 + 0.02T_{opt}(x) - 0.0005[T_{opt}(x)] \tag{7}$$

$$T_{\varepsilon 2}(x,t) = \frac{1.1814}{\left\{1 + e^{0.2 \times [T_{opt}(x) - 10 - T(x,t)]}\right\} \times \left\{1 + e^{0.3 \times [-T_{opt}(x) - 10 + T(x,t)]}\right\}} \tag{8}$$

$$W_{\varepsilon}(x,t) = 0.5 + 0.5 \times \frac{ET(x,t)}{PET(x,t)} \tag{9}$$

where $T_{\varepsilon 1}(x,t)$ and $T_{\varepsilon 2}(x,t)$ are the temperature stress factors of the light use efficiency of pixel $x$ at time $t$; $W_{\varepsilon}(x,t)$ represents the water stress factor of the light use efficiency of pixel $x$ at time $t$; and $\varepsilon_{\max}$ [gC·MJ$^{-1}$] is the maximum light use efficiency, which differs with land-cover type (Running et al., 2000, Table 3). $T(x,t)$ (°C) is the average temperature of pixel $x$ at time $t$, and $T_{opt}(x)$ (°C) is the temperature at the highest NDVI in a year. $ET(x,t)$ (mm) and $PET(x,t)$ (mm) are actual evapotranspiration and potential evapotranspiration from the MODIS ET/PET dataset.

**Table 2.** Maximum and minimum RVI of different vegetation types.

| Vegetation types | RVI$_{max}$ | RVI$_{min}$ |
|---|---|---|
| Deciduous needle-leaf forest | 6.63 | 1.05 |
| Deciduous broad-leaf forest | 6.91 | 1.05 |
| Sparse woods | 4.49 | 1.05 |
| Steppe | 4.46 | 1.05 |
| Urban lands | 4.46 | 1.05 |
| Desert | 4.46 | 1.05 |
| Croplands | 4.46 | 1.05 |

**Table 3.** Maximum light use efficiency of different vegetation types.

| Vegetation Types | $\varepsilon_{max}$ [gC·MJ$^{-1}$] |
|---|---|
| Deciduous needle-leaf forest | 1.008 |
| Deciduous broad-leaf forest | 1.259 |
| Sparse woods | 0.774 |
| Steppe | 0.608 |
| Croplands | 0.604 |
| Other | 0.389 |

### 3.2. Model Optimization and Validation

We optimized the CASA model to accurately simulate the net primary productivity (NPP). In the CASA model, previous research has always simulated evapotranspiration based on meteorological data. MODIS ET datasets provide high-resolution evapotranspiration data based on satellite data and surface meteorological data, which is a better choice for the calculation of the water stress factor of the light use efficiency. The MODIS land cover product from 2001 to 2020 at yearly temporal frequency was used to determine the maximum light use efficiency and maximum RVI for each pixel, which allowed estimations of NPP to reflect land use changes over the past 20 years. The continuous changes in land use obtained from this product made the model estimation more reasonable. In this study, we simulated monthly NPP between 2001 and 2020 on the LP. Observed biomasses in 2010 were converted to obtain the actual NPP in the sampling area, which were used to assess the CASA model. As illustrated in Figure 2, the proximity between observed and modeled

NPP were characterized by NRMSE = 15%, $R^2$ = 0.92, which indicated that NPP on the LP could be accurately simulated by the CASA model.

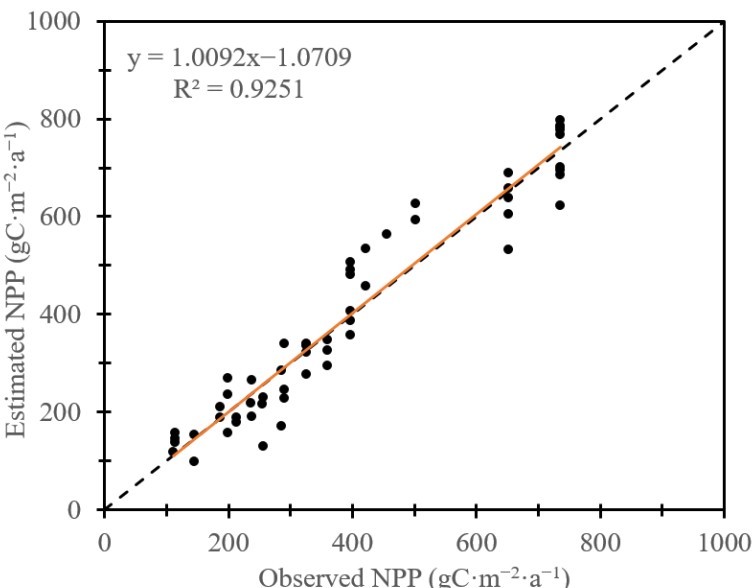

**Figure 2.** Observation vs. simulation based on the CASA model over our study area in 2010. $R^2$ = 0.92; NRMSE = 0.15.

## 4. Results

### 4.1. Spatiotemporal Pattern of NPP

The NPP patterns on the LP varied along with the differences in the land cover and precipitation gradient [50]. In other words, the NPP decreased from the southeast forest to the northwest desert, varying along the rainfall gradient. Meanwhile, the relatively low NPP distribution in the southeast and relatively high NPP over the northwest were produced by croplands. According to the comparison of NPP distribution in 2001 and 2020, a huge improvement in the NPP has occurred in the past 20 years (Figure 3). NPP increased by 165.2% from 129.1 gC·m$^{-2}$·a$^{-1}$ to 294.3 gC·m$^{-2}$·a$^{-1}$ over the study area. Low-NPP regions are shrinking in the northwest of the LP, while high-NPP areas are expanding in the southeast of the LP. In sub-regions of forest–steppe, NPP in the TDF increased by 108.4% from 279.8 gC·m$^{-2}$·a$^{-1}$ to 583.1 gC·m$^{-2}$·a$^{-1}$, while NPP in the TFS increased by 150.3% from 189.1 gC·m$^{-2}$·a$^{-1}$ to 473.4 gC·m$^{-2}$·a$^{-1}$. In the steppe region, NPP in the TTS increased by 145.0% from 65.5 gC·m$^{-2}$·a$^{-1}$ to 160.4 gC·m$^{-2}$·a$^{-1}$. In the desert–steppe regions, NPP in the TDS increased by 32.6% from 25.0 gC·m$^{-2}$·a$^{-1}$ to 33.1 gC·m$^{-2}$·a$^{-1}$. NPP in the AFS increased by 28.6% from 192.2 gC·m$^{-2}$·a$^{-1}$ to 247.1 gC·m$^{-2}$·a$^{-1}$. NPP in the TSD increased by 56.7% from 53.6 gC·m$^{-2}$·a$^{-1}$ to 83.9 gC·m$^{-2}$·a$^{-1}$. In general, improvement of NPP in the forest and steppe was more obvious than that in the desert–steppe.

Significant continuous rises (an increase of 7.45 gC·m$^{-2}$·a$^{-2}$, $R^2$ = 0.9328, $p < 0.01$) in annual NPP have occurred on the LP during the past 20 years (Figure 4). At the pixel scale, the increases in NPP appear heterogeneous due to the difference in land cover on the LP. Most of pixels (93.4%) on the LP showed increases in NPP, and strong and significant ($p < 0.05$) increases of NPP occurred in 73.6% of the LP, mainly distributed in the middle and southeastern part of the plateau. NPP in forest areas showed higher rises than NPP in non-forest areas. According to the comparison of the NPP variation and land cover maps, NPP of forests (green pixels in the left panel of Figure 4) increased by more than 10 gC·m$^{-2}$·a$^{-2}$, while NPP data from non-forest regions (yellow or red pixels) indicate lower increases or faint decreases over the past 20 years. In the forest–steppe sub-regions, NPP in TFS increased significantly by 13.35 gC·m$^{-2}$·a$^{-2}$ ($p < 0.05$), while NPP in TDF regions showed lower rises (12.81 gC·m$^{-2}$·a$^{-2}$) due to the decreases of NPP in the

croplands in TDF areas. In the steppe region, NPP in TTS showed significant increases at a rate of 5.27 gC·m$^{-2}$·a$^{-2}$ ($p < 0.05$). In the desert–steppe regions, NPP saw faint increases in the TSD (2.55 gC·m$^{-2}$·a$^{-2}$), AFS (1.89 gC·m$^{-2}$·a$^{-2}$), and TDS (1.61 gC·m$^{-2}$·a$^{-2}$) regions. In general, the rapid growth of NPP in forest regions led significant continuous rises on the LP over the past 20 years.

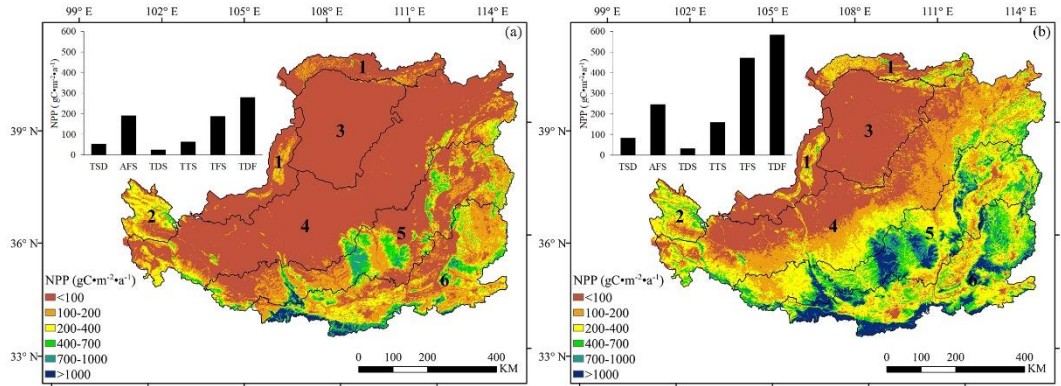

**Figure 3.** NPP distribution in (**a**) 2001 and (**b**) 2020 on the Loess plateau. The numbers on the pixels represent six sub-regions on the LP: (1) temperate steppe desert (TSD), (2) alpine forest–steppe (AFS), (3) temperate desert–steppe (TDS), (4) temperate typical steppe (TTS), (5) temperate forest–steppe (TFS), (6) temperate deciduous forest (TDF). Bar charts show the average NPP in six sub-regions in 2001 and 2020.

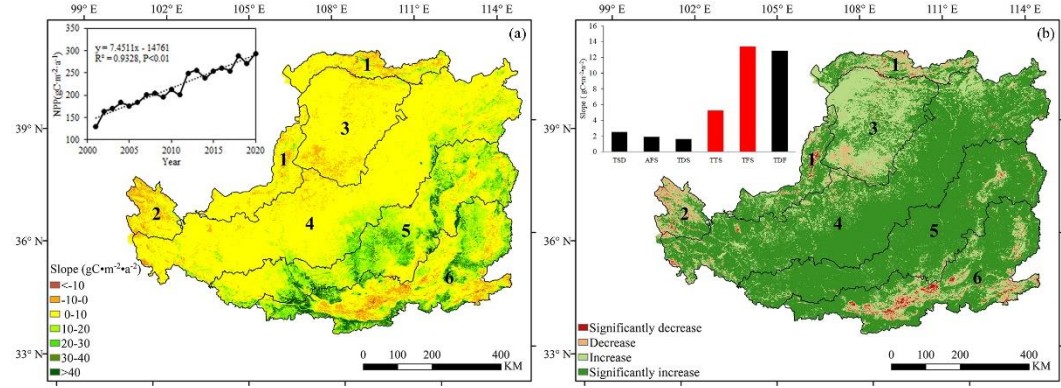

**Figure 4.** (**a**) Spatial patterns of variations of NPP on the LP during 2001−2020. The numbers on the pixels represent six sub-regions on the LP: (1) Temperate steppe desert (TSD), (2) alpine forest–steppe (AFS), (3) temperate desert–steppe (TDS), (4) temperate typical steppe (TTS), (5) temperate forest–steppe (TFS), (6) temperate deciduous forest (TDF). The line chart shows the trend in average NPP over the study area. (**b**) Significance of variations of NPP on the LP during 2001−2020. The red bar in the right panel indicates a significant increase ($p < 0.01$) in average NPP over the TTS and the TFS regions.

### 4.2. Quantitative Analysis of Contributions of Driving Factors on Variations in NPP

Climate variations and human land use management were both found to be important driving factors of vegetation change on the LP [36]. According to our contribution analysis, changes in regional precipitation, temperature, and solar radiation made positive contributions to NPP increases. Precipitation contributed 1.53 gC·m$^{-2}$·a$^{-2}$ (20.5%) of the NPP increases, whereas temperature and solar radiation only contributed 0.59 gC·m$^{-2}$·a$^{-2}$ (7.9%) and 0.55 gC·m$^{-2}$·a$^{-2}$ (7.4%), respectively. In addition, the other factors (mainly human land-use management) contributed 4.78 gC·m$^{-2}$·a$^{-2}$ (64.2%) to the NPP increases on the LP. At the pixel scale, the negative contributions (red pixels in Figure 5) and low positive contributions (yellow pixels in Figure 5) were considered invalid contributions. According to the map of contributions (Figure 5), heterogeneities of contributions were different for each factor. Precipitation indicated obvious positive contributions on the

southwestern and northeastern mountain forests, which mainly distributed in TFS and TTS. Temperature showed positive contributions on the eastern part of the LP, while radiation showed positive contributions on the southwest of the LP. Human land-use management indicated huge contributions to NPP increases in the southeastern forest–steppe and obvious contributions in the middle steppe on the LP. In general, human activities on the LP appear to have made a greater contribution to the NPP increases than climate variations.

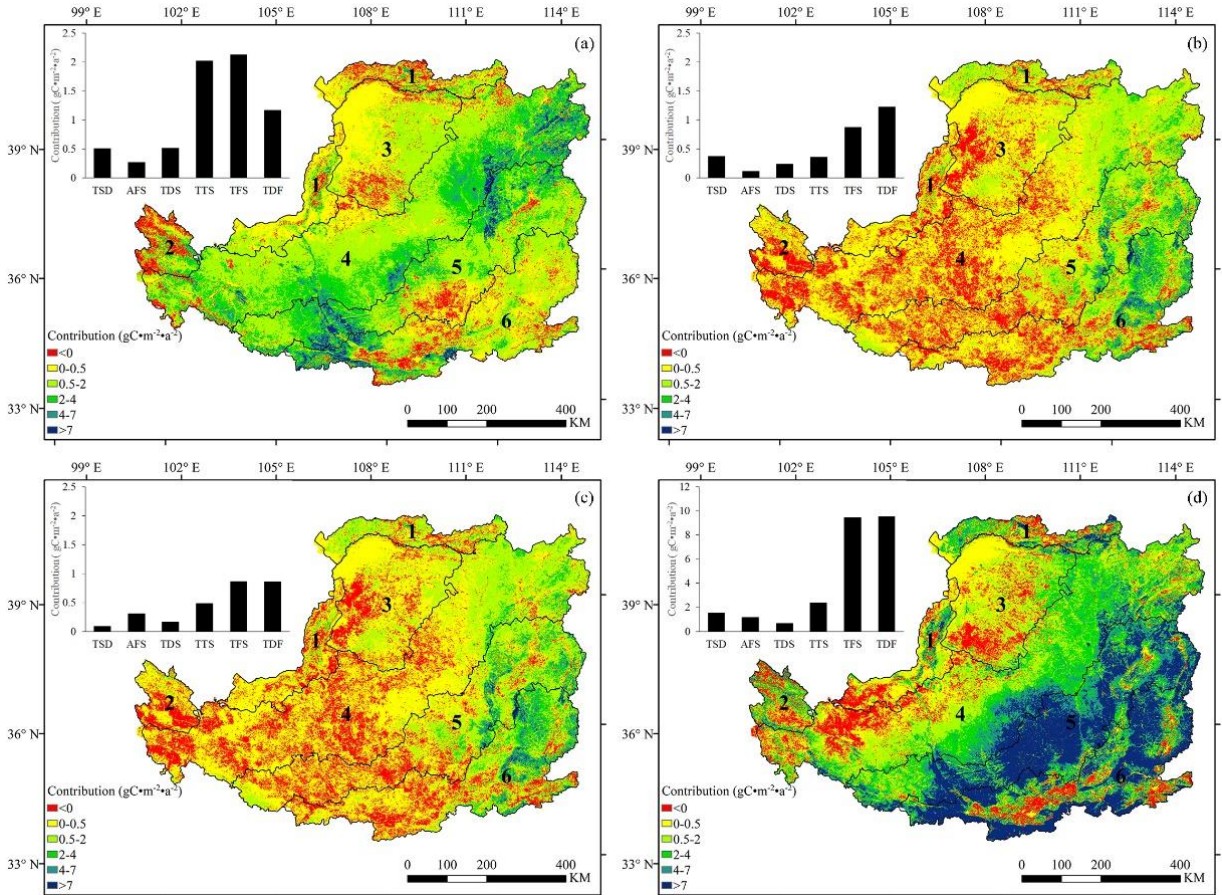

**Figure 5.** Contribution of each driving factor to interannual variations in NPP: (**a**) precipitation; (**b**) temperature; (**c**) radiation; (**d**) human land use. Red pixels represented weakly negative contributions to NPP increases. Yellow pixels represented faint positive contributions to NPP increases. Green pixels represented obvious positive contributions to NPP increases. The numbers on the pixels represent six sub-regions on the LP: (1) Temperate steppe desert (TSD), (2) alpine forest–steppe (AFS), (3) temperate desert–steppe (TDS), (4) temperate typical steppe (TTS), (5) temperate forest–steppe (TFS), (6) temperate deciduous forest (TDF).

The driving forces behind vegetation change were heterogeneous along with spatial patterns of the land cover on the LP. Based on a comprehensive analysis of contributions of each driving factor in Figure 5, human land use dominated the NPP variations in 58.8% of areas on the LP, which were mainly distributed in the TDF, TFS, central TTS, and TSD regions. Climate-dominant areas were mainly distributed in the TTS and TDS regions (Figure 6). In the forest–steppe regions (TDF and TFS), more than 80% of pixels indicated human activities dominated NPP variations in the forests and croplands. In the TSD and AFS, increases in NPP were dominated by human activities in more than 50% of pixels, including many areas of cropland. In the steppe regions (TTS and TDS), increases in NPP in the grasslands were mainly influenced by climate variation, while precipitation was the most important climate factor for grasslands. In general, NPP in forests and croplands were dominated by human activities, while grassland NPP was mainly influenced by climate variation on the LP over the past 20 years.

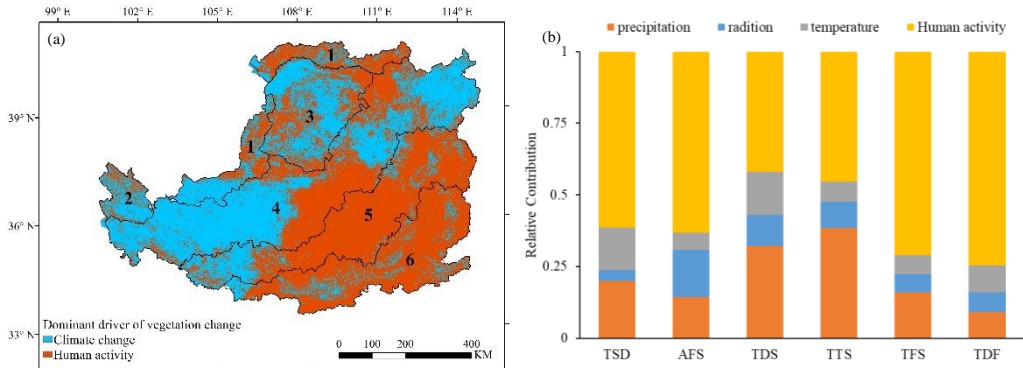

**Figure 6.** (**a**) Spatial pattern of dominant driver of variations of NPP on the LP. The numbers on the pixels represent six sub-regions on the LP: (1) Temperate steppe desert (TSD), (2) alpine forest–steppe (AFS), (3) temperate desert–steppe (TDS), (4) temperate typical steppe (TTS), (5) temperate forest–steppe (TFS), (6) temperate deciduous forest (TDF). (**b**) Relative contribution of climate factors and human activity to variations of NPP in sub-regions on the LP.

Vegetation growth was determined by the combined effects of driving factors. Therefore, analysing these interactive effects has been an important part of deepening our understanding of the attribution of vegetation variations. Based on Geo-detector analysis, interactive effects were calculated to characterize the weights and interactive effects of different driving factors on vegetation variations. In Figure 7, all of the effects of the interaction between land use and climate factors exceeded 0.5 in the TTS, TFS, and TDF sub-regions, which indicates that land use was a core driver of vegetation growth in the southeastern LP. In the northwestern regions, the interactive effects of land use∩climate factors were relatively weak, which meant that the impacts of land use were weaker, and climate factors had a higher impact weight than that in the southeastern LP. Furthermore, the interaction between land use and precipitation had the highest ability to explain changes in NPP in the TDS, TTS, TFS, and TDF sub-regions, which meant land use and precipitation were the two most important factors for vegetation variation in most areas of the LP. In general, vegetation was mainly affected by land use in the southeastern LP, while the influence weights of climatic factors increased in the northwestern part.

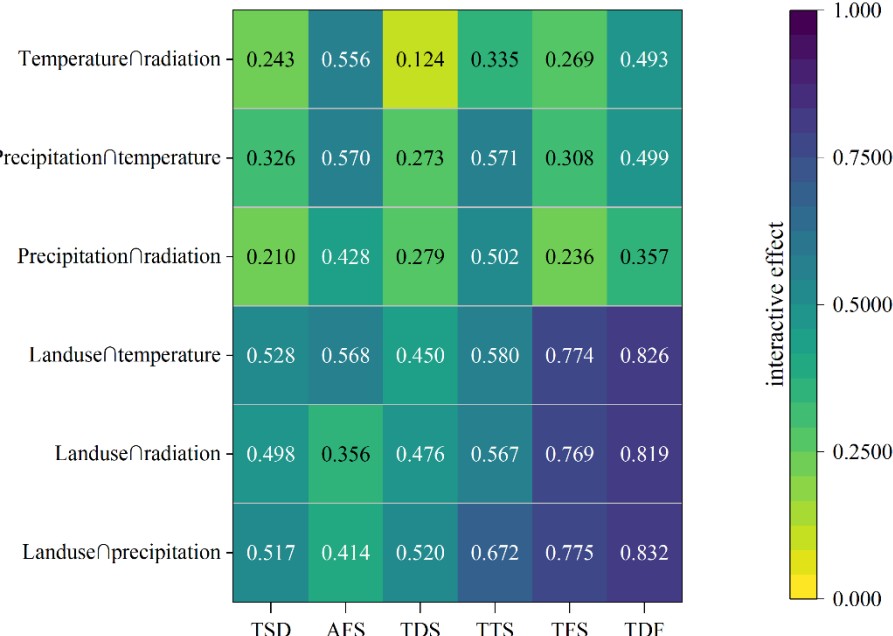

**Figure 7.** The interactive effects of climate factors and land use on the NPP on the LP.

### 4.3. Changes in Population Patterns and Vegetation Types on the LP

GTGP, the most important land-use management program in China, has made great contributions to the greening of the LP. Eco-migration has been a very important aid to support the GTGP's implementation. Panels (a) and (b) in Figure 8 represent the map of population density on the LP in 2000 and 2020, divided into different levels. In this study, we defined the population density as follows: $0-5/km^2$—areas of extremely low density, depopulated zone; $5–500/km^2$—areas of moderate density, rural area; $>500/km^2$—areas of high density, urban area. Depopulated zones increased by 63.86% from 15.87% to 26.00% of the whole region from 2001 to 2020. Rural areas decreased by 15.64% from 81.10% to 68.41% of the LP, while urban areas increased by 84.06% from 3.04% to 5.59% of the LP. By comparing Figures 1 and 8, it can be seen that the newly added depopulated zones (red ellipses) were mainly distributed in the forest areas (TDF and TFS), which indicates that the government has protected and restored forests through eco-migration over the past 20 years, and that this has led the significant continuous rises in NPP on the LP over the past 20 years. Combined with the previous results, ecological migration has played an important role in afforestation/reforestation, which was conducive to the rapid growth of NPP in forest regions.

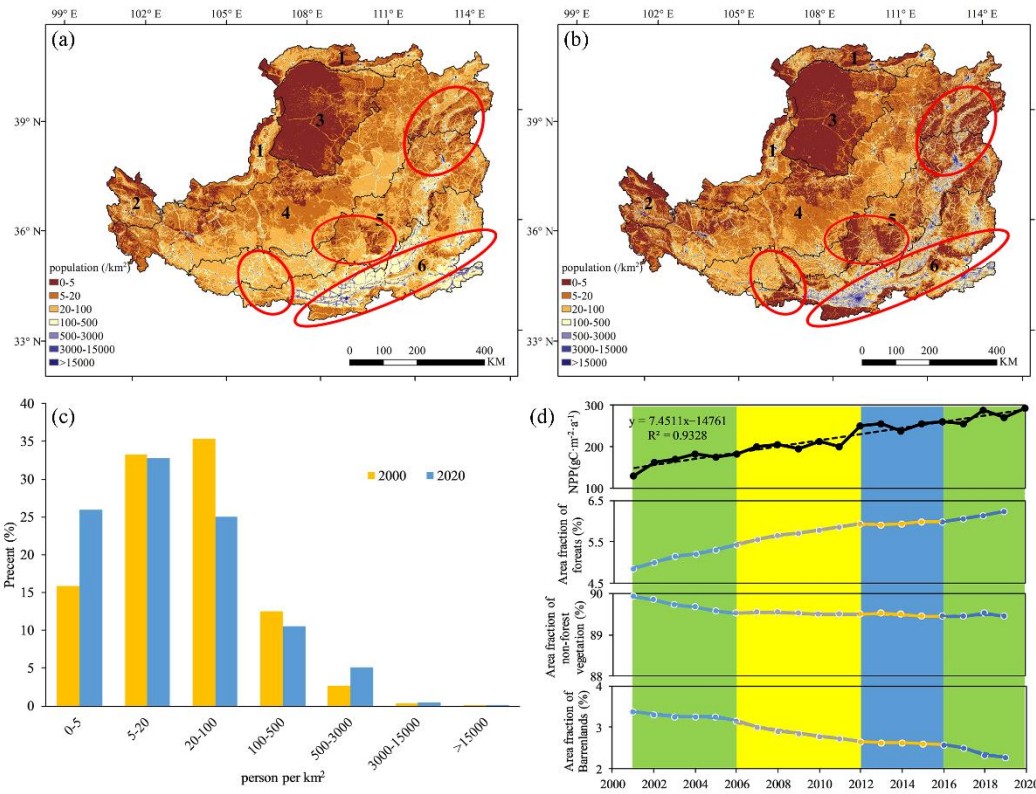

**Figure 8.** Human activity on the LP over the past 20 years. Panels (**a**) and (**b**) are maps of population density on the LP in 2000 and 2020, respectively. Panel (**c**) shows fractions of pixels for different population densities. Panel (**d**) represents variations in NPP with percentage area of forests, non-forest vegetation, and barren land on the LP. The red ellipses in panel (**a**) and (**b**) represent the regions with extreme decreases in population density. The color bars in panel (**d**) show the various stages of land use-managements.

According to land-cover change, the land-cover map was reclassified into three types: forest, non-forest vegetation, and barren land. The forested area increased by 28.7%, while barren land decreased by 32.8%. Based on the white paper, "Twenty Years of Converting Farmland to Forest and Grass in China", changes in forests, non-forest vegetation, and barren land can be divided into four stages: 2001–2006; 2006–2012; 2012–2016; and 2016–2019. In first stage, forest areas continually increased, while non-forest vegetated

areas continually decreased and barren land weakly decreased on the LP. The NPP on the LP increased significantly in 2001–2006. In stage 2, forest areas still continually increased, while barren lands tangibly decreased. However, non-forest vegetated areas underwent almost no change. The NPP on the LP increased significantly in 2006–2012. In stage 3, all the forests, non-forest vegetation, and barren lands underwent no obvious change. The NPP increases were very weak in this stage. In stage 4, forest areas increased, while barren lands tangibly decreased. Non-forest vegetated areas underwent almost no change. NPP increased after 2016 on the LP. In general, increases in forested areas have led to the increases in NPP on the LP over the past 20 years.

*4.4. Impact of Strong El Niño Event on the Vegetation on the LP*

Precipitation gradients determined the spatial distributions of vegetation types on the LP, which affected the spatial patterns of NPP. In recent decades, extreme events of atmospheric oscillations, such as ENSO events, have had important impacts on global climate change [51]. In the north of China, precipitation has always been low in El Niño years and high during La Niña events [42], which affects vegetation variations in arid and semi-arid regions. Essentially, La Niña event precipitation leads to positive influences on NPP, while El Niño events have negative effects on vegetation on the LP. According to the sea-surface temperature anomaly (SSTa) for Niño 3.4 (Figure 9), six El Niño events and seven La Niña events have occurred during the past 20 years. According to the comparison of the NPP, SSTa, and precipitation data, the super El Niño event in 2015 led to a decrease in water vapor transport from the ocean to the LP (Figure 9d,e), which caused the lowest precipitation (29.5% lower than multi-year average precipitation) in the growing season over the past 20 years. The extreme drought eventually led to the vegetation browning, and then made 2015–2017 a low point in the greening of the Loess Plateau.

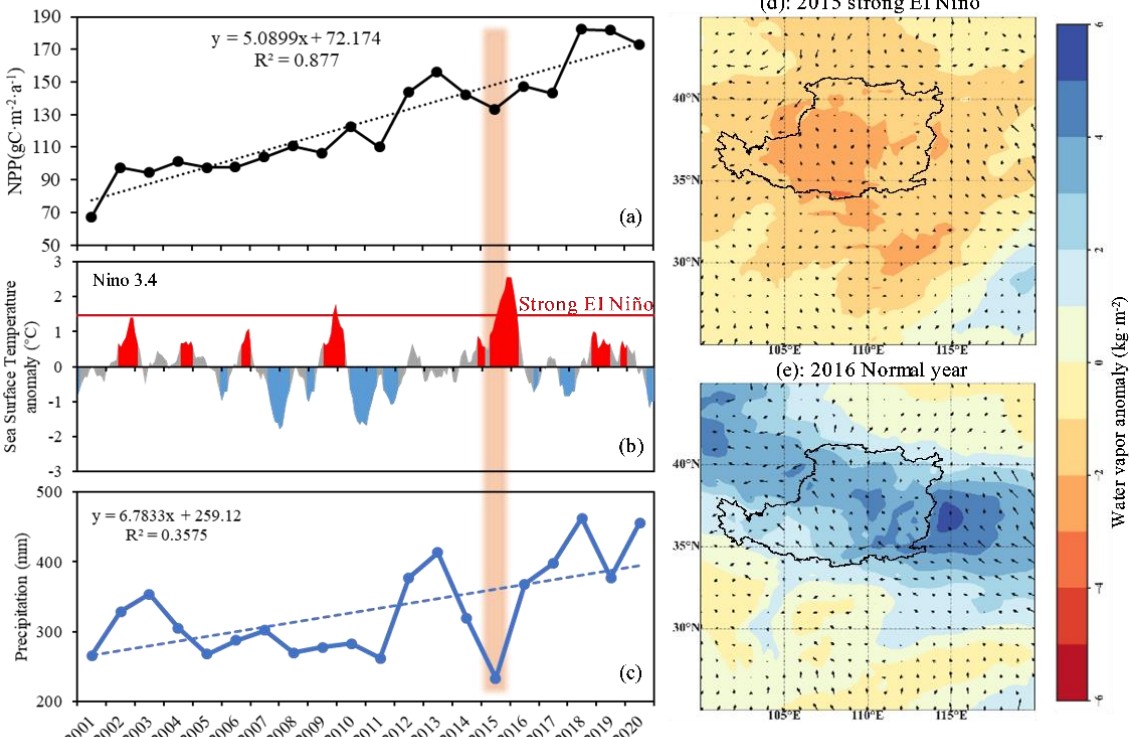

**Figure 9.** Time series of (**a**) NPP, (**b**) SSTa for Niño 3.4, and (**c**) precipitation in growing season over the periods 2001–2020 and anomaly distribution of vertically integrated summer water vapor in (**d**) 2015 and (**e**) 2016. The brown bar represents the decrease in NPP and precipitation affected by the strong El Niño event in 2015. The arrows in panel (**c**) and (**d**) represent the wind direction.

## 5. Discussion

### 5.1. Heterogeneity of Vegetation Variation over the LP

Complicated topography, surface runoff, and precipitation gradients determine the spatial distributions of vegetation types on the LP, which affects the spatial patterns of NPP. According to the elevation map of the LP, there are many mountains in the southern and eastern part of the LP. These mountains receive a lot of water vapor from the East Asian monsoon and block the transportation of water vapor to the northwestern region. Therefore, obvious precipitation gradients are formed on the LP, which leads to the spatial patterns of "forest–steppe–desert steppe" from southeast to northwest. Meanwhile, surface runoff also has impacts on the vegetation type. As can be seen on Figure 1, the Yellow River flows through the northern steppe desert of the LP, providing enough water for vegetation growth. After more than 2000 years of water conservancy and agricultural developments, the Hetao plain was formed in the arid desert area of the LP, becoming an important grain-producing area of China [52]. In this study, it was found that NPP varied with changes in land-cover type. The highest NPP occurred in mountain forests, followed by cropland, steppe, and desert steppe. Over the past 20 years, regional asymmetrical vegetation improvement has occurred on the LP. The improvements in the southeastern forest and forest–steppe were more obvious than those in the northwestern desert–steppe.

Continuous increases in vegetation productivity have been heterogeneous over the past 20 years on the LP. NPP in most regions has undergone significant increases, mainly distributed in forest and steppe areas. The forests which saw a rapid growth in NPP are mainly distributed in the southeast regions where there were many mountains, including the Qinling Mountains, Lvliang Mountains, and Taihang Mountains. Mountain forests were found to have rich biological species, more hierarchical structures, and higher photosynthetic productivity, giving them the benefits of regulating climate, conserving water and soil sources, acting as windbreaks, and assisting with sand fixation. The improvements of forest vegetation were of great significance to the ecological restoration of the LP. NPP in typical steppe areas also indicated significant increases, mainly distributed in the central areas of the LP. The improvements in steppe vegetation were of great significance to the control of desertification on the LP. Meanwhile, urbanization has caused decreases in NPP in the Hetao plain and Guanzhong plain.

### 5.2. Heterogeneity of Attribution of the Vegetation Variations

The GTGP is China's largest ecological engineering program, and was initiated in 1999, mainly focusing on mountainous regions [16]. The LP was a key area for the implementation of the project. In this study, the contributions of human activities and climate variations on NPP were 64.2% and 35.8%, respectively. Increases in newly added depopulated zones and forest areas indicate that eco-migration and afforestation/reforestation were core drivers of increased NPP in forest areas (TDF and TFS). Human activities were the dominant factor driving increases in NPP in the southeastern LP, while climate variations were the dominant factor driving increases in NPP in the west and north of the LP.

The temporal stages in changes to forest and non-forest areas were related to the stages of the GTGP. In the initial stage of the project, 2001–2006, forested areas increased significantly, while non-forest areas decreased tangibly. In the consolidation phase, from 2006 to 2012, the conversion of cultivated land to forests was suspended in order to ensure sufficient cultivated land area, while the afforestation of barren mountain and the setting aside of hills for afforestation continued. The forested areas continually increased, while the barren lands tangibly decreased. In the transition phase, from 2012 to 2016, the government investigated and studied the implementation plan for the next phase of GTGP. All the forest, non-forest, and barren areas underwent no obvious change. In 2016, the second stage of GTGP started. Forested areas increased, while barren land tangibly decreased. According to Figure 8, the phases of increasing NPP corresponded to the changes in forest areas over the past 20 years, which indicates that the mountain forests in the southeast of the LP were key project areas for the GTGP. Meanwhile, ecological migration was

conducive to reducing the negative effects of human activities in the forest areas. In general, afforestation/reforestation had a dominant effect on increases in NPP in the southeastern LP. In the western and northern areas of the LP, the increases in precipitation had a dominant effect vegetation growth on the steppe in semi-arid regions.

In this study, we mainly focused on the influence of climate change and human activities on the NPP on the LP, and gave less consideration to other factors, such as $CO_2$ fertilization effects and nutrient limitation. In the future, more comprehensive consideration should be given to the driving mechanisms of changes to the NPP of vegetation. Meanwhile, extreme atmospheric oscillations such as ENSO events have had important impacts on global climate change in recent decades. This paper preliminarily explored the impact of strong ENSO events on NPP. We found that strong El Niño events exacerbated obstruction of the GTGP's goals. The influence of atmospheric oscillations on the carbon cycle requires further analysis in the future.

## 6. Conclusions

NPP has undergone significant continuous rises (7.45 gC·m$^{-2}$·a$^{-2}$, $R^2 = 0.9328$, $p < 0.01$) on the LP over the past 20 years, with rapid growth of NPP in forests in the southeastern regions a key driver of this. The contributions of human activities and climate variations to NPP were found to be 64.2% and 35.8%, respectively. The effects were heterogeneous, varying with spatial variations in land cover. Mountain forests in the southeast of the LP were key project areas for the GTGP. In addition, ecological migration and the population agglomeration caused by urbanization were conducive to reducing the negative effects of human activities in forested areas. As a result, afforestation/reforestation had a dominant effect on increases in NPP in the southeastern LP, whereas in the western and northern areas, climate variations, especially increases in precipitation, dominated vegetation growth on the steppe in semi-arid regions. Furthermore, strong ENSO events exacerbated obstruction of the GTGP's goals. In general, the results of this study provide a comprehensive perspective on the detection and attribution of greening in a core GTGP area of China. In future research, climate extremes caused by ENSO should be identified to study and predict vegetation variations on the LP.

**Author Contributions:** Conceptualization, W.G.; methodology, W.G. and X.N.; software, X.N. and W.G.; formal analysis, X.N. and W.G.; validation, X.N. and X.L.; writing—original draft preparation, X.N. and W.G.; writing—review and editing, X.N., X.L. and S.L.; supervision, W.G. and S.L.; project administration, W.G. and X.N.; funding acquisition, W.G. All authors have read and agreed to the published version of the manuscript.

**Funding:** This research was funded by the Strategic Priority Research Program of Chinese Academy of Sciences (XDB40020200), the National Natural Science Foundation of China (42071124), the Carbon Neutrality Monitoring Decision Application System (ZY-2021-005), the State Key Laboratory of Loess and Quaternary Geology (SKLLQG1809), and the Fundamental Research Funds for the Central Universities (XZY012019008).

**Data Availability Statement:** Publicly available datasets were used in this study. The MODIS products were acquired from the Land Processes Distributed Active Archive Center (LP DAAC) (https://lpdaac.usgs.gov (accessed on 1 June 2021)). GPM IMERG and FLDAS datasets were publicly available from the NASA Goddard Earth Sciences (GES) Data and Information Services Center (DISC) (https://disc.gsfc.nasa.gov (accessed on 1 June 2021)). Multivariate ENSO Index was available from the NOAA Physical Sciences Laboratory (https://psl.noaa.gov/enso/ (accessed on 1 June 2021)). ERA5 monthly averaged data on single levels was available from the European Centre for Medium-Range Weather Forecasts (ECMWF) (https://cds.climate.copernicus.eu/ (accessed on 1 June 2022)).

**Conflicts of Interest:** The authors declare no conflict of interest. The funders had no role in the design of the study; in the collection, analyses, or interpretation of data; in the writing of the manuscript, or in the decision to publish the results.

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
