# Peer review of "Heterogeneity of Increases in Net Primary Production under Intensified Human Activity and Climate Variability on the Loess Plateau of China"

_remotesensing, doi:10.3390/rs14194706_

Round 1

Reviewer 1 Report

The authors studied heterogeneity of increases in net primary production (NPP) on the Loess Plateau (LP) from 2001 to 2020. NPP was one of the key parameters that reflect the production capacity of plant communities under various natural environmental conditions. They accurately simulated the NPP on the LP after model optimization. Then they applied the NPP to study the spatial-temporal patterns of vegetation variations, disentangle the influence of climate change and human activities, and discuss the impacts of atmospheric oscillation on vegetation NPP on the LP. They found that human activities contributed 64.2% on the NPP increases, while climate variations contributed 35.8%. Meanwhile, they also revealed that strong El Niño event exacerbated obstruction on large-scale ecological restoration. These conclusions were very useful for the carbon cycle assessment and sustainable development evaluation.

I recommend publication with minor corrections.

some minor comments below:

L35. “Monitoring and quantifying changes in carbon sinks in mid-high latitudes, therefore, is essential for understanding the global carbon cycle and future climate change.” Please replace “is” with “are”

L37. “A great greening trend of the globe has been observed during the past four decades”, Please replace “great greening trend of the globe” with “great global greening trend”.

L175. Section 2.3. Does this contribution only represent the NPP changes between 2000 and 2020? Does the contribution change with the target year? Say like between 2000 and 2011 or 2015, due to an obvious decline of NPP shown in the Figure 8.

L316. Section 4.3. I think the explanation for population patterns is insufficient to support a clear understanding of results, especially figure 7 and its associated discussions.

Figure.6. Revise the labels

Figure.8. This figure shows the anomaly distribution of vertically integrated water vapor. Please specify data source. Besides, please also explain the meaning of the arrows in the figure.

Reviewer 2 Report

This manuscript (MS. No. remotesensing-1894337) investigated detection and attribution of increases in net primary production (NPP) on the Loess Plateau (LP) from 2001 to 2020. They found some interesting results in this study. For example, they revealed that rapid growths of forest NPP and expansion of forest area in the southeastern regions led vegetation restoration of LP. They also attempted to provide some reasonable explanations at last. The conclusions of this study may help us understand how human activity and climate variability may heterogeneously affect the vegetation productivity over this region. Overall, this is a well-organized paper, the methodology is reasonable, and the major conclusions are supported by the results. This study matches with the scope of the journal. I have some comments below which may help the authors further improve it. I suggest a minor revision is needed before consideration for acceptance by the journal of “Remote Sensing”.

Detailed comments :

L259. “non-forest areas” or “non-forest regions”.

L382. I consider here a lack of informative discussions on why TFS and TDF were dominated by human influence.

L407. How do you consider the ENSO in your way to disentangle the effects of human and climate change on NPP?

Figure.1.  Add “(a)” and “(b)” in the figure caption

Figure.3.  figure caption:” NPP distribution in 2001 (left panel) and 2020 (right panel) on the Loess plateau.” Please replace ” left panel; right panel” with “a; b”

Figure.7. This paper investigated NPP dynamic during 2001-2020. Here you used population density in 2000 and 2018 to indicate the human activity. Please use population density in 2020.

In addition, I also suggest adjust Font size in figures, such as figure 1, figure 7, and figure 8.

Reviewer 3 Report

Accept after minor revision, comments in file attached.

Reviewer 4 Report

This manuscript analyzed the spatial and temporal variations in net primary productivity (NPP) over the Loess Plateau, China, during the 2001-2020 period. It also attributed the changes in NPP to different driving factors, including temperature, precipitation, solar radiation and human activities. Overall, this manuscript falls within the scope of Remote Sensing. However, there are some issues in the methodology and dataset part, which need to be improved. In addition, I would suggest that the authors get editing help from someone with full professional proficiency in English, as the current manuscript has substantial language issues.

Major concerns:

1. This manuscript used the partial derivative method to quantify the relative contribution of different driving factors in explaining the spatiotemporal variations of NPP. However, the method used to implement such analysis only considers the direct impact (i.e., individual impact) of these factors on NPP, and the impacts of the interactions between these factors on NPP are not considered. Ignoring the interactions between these factors may influence the robustness of the conclusions. Therefore, I suggest the authors adopt some advanced methods to conduct the attribution analysis. In addition, the equation (1) does not include human activities, and the authors may need to further check this equation.

2. The authors used temperature and radiation datasets from FLDAS, and precipitation datasets from GPM_3IMERGM to conduct their analysis. I am very curious why the authors did not use climate variables from the same data sources for analysis (e.g., extract temperature, precipitation, and solar radiation from FLDAS). Since there are large uncertainties and discrepancies between climate variables from different sources, the “multi-source” climate datasets used here may introduce large uncertainties to their results and conclusions.

Besides, I also have some specific comments below.

Line 30: Awkward sentence: what do you mean by “As the source and sink of carbon dioxide in the atmosphere”

Line 62-63: Inaccurate descriptions. There are many reconstructed land use and land cover (LULC) datasets available to drive models, such as the Land-Use Harmonization 2 dataset (LUH2).

Line 65-67: I would not say it is an innovation of your study because you used the LULC product from MODIS to drive your model. In addition, the MODIS products also have large uncertainties, do you conduct any site-scale validation for the MODIS’s LULC dataset?

Line 80-82: Awkward sentences.

Table 1: Annual temperature/precipitation? During which period?

Line 147: first refined……, then what?

Line 225-226: What do you mean by “assess the validation…”

Line 227: What is DASFs?

Figure 3 and 4: You should include the full name of the abbreviations (e.g., TSD and AFS) in the figure caption.

Figure 7: The quality of the figures should be greatly improved.

Round 2

Reviewer 4 Report

The authors have made efforts to address my concerns. I recommend accepting this manuscript in its current form.